# The Impact of Underlying Opaque White Coating Parameters on Flexographic Print Quality

**Renata Tomerlin** [1], **Dean Valdec** [2,*], **Mario Tomiša** [2] and **Damir Vusić** [3]

1   Podravka d.d., Ante Starčevića 32, 48000 Koprivnica, Croatia; renata.tomerlin@podravka.hr
2   Department of Art Studies, University North, Trg dr. Žarka Dolinara 1, 48000 Koprivnica, Croatia; mtomisa@unin.hr
3   Department of Multimedia, University North, Trg dr. Žarka Dolinara 1, 48000 Koprivnica, Croatia; dvusic@unin.hr
*   Correspondence: dean.valdec@unin.hr

**Abstract:** Opaque white ink is highly important when printing on transparent substrates. The purpose of the white ink is to completely or partially cover the content of the packaging as well as to ensure the printing of other colors over it according to the appropriate printing specifications. The main goal of this study was to research the impact of anilox roller volume of opaque white on its opacity, and, consequently, on opacity, as well as the CIELAB values of colors that are printed over it. The research was conducted in three printing stages, during which the volume of the anilox roller was increased in linear steps of 3 cm$^3$/m$^2$ while other parameters remained constant. The analysis of white ink covering properties was carried out on microscopic print images using ImageJ software. The results showed that a linear increase in the anilox roller volume resulted in an exponential change in white ink opacity as well as the opacity of color samples. It was also determined that a volume increase of 6 cm$^3$/m$^2$ was required to visually notice color difference. Namely, medium lightness colors more strongly reacted to changes in white ink opacity compared to light and dark colors.

**Keywords:** flexographic print quality; opaque white ink; covering property; anilox volume

## 1. Introduction

Achieving a high level of white ink opacity in printing presents a great challenge for flexographic printing industry today. The white ink printed on transparent substrates due to lack of opacity and nonuniform ink layer reflects less light, resulting in dirty appearance. Multicolor graphics printed on such substrate look blurry, and the colors are less saturated and not faithful. Thus, coverage property (CP) is the most important performance of white ink [1], and the main factor that has the greatest influence on CP is ink dispersion [2]. In order to improve characteristics of white color, inks with a higher viscosity, and anilox rollers with a larger volume, are used and printing with a reduced printing speed or printing with two printing plates is carried out [3]. The question is how much opacity can be achieved with high opaque white ink on transparent materials. It is even more important to determine the optimal value of white ink opacity. In today's flexographic printing environment, opacity above 60% can easily be achieved, as there have been huge advances in all the necessary factors that affect the opacity value, including anilox rollers, inks, doctor blade, mounting adhesive tape, printing plate properties, nip pressure, printing speed, and substrate treatment [4,5].

The desired level of white ink opacity is determined by the thickness of the layer, which is directly related to the volume of the anilox roller measured in BCM (billions of cubic micrometers per square inch). There are two types of anilox rollers according to their basic purpose, namely printing and coating [6]. Anilox rollers for partial printing have a higher line screen, which means that they have smaller cell openings, as well as a smaller volume. Typical anilox printing rollers have a volume of up to 8 BCM; its final selection depends

on the type of work being printed (solids, text, vignettes, or multicolor illustrations) [7]. Anilox rollers intended for coating (complete coloring of the material) have a smaller line screen, meaning relatively thicker cell walls, larger cell openings, and a larger cell volume. Typical anilox coating rollers have a volume of 8 BCM or more. Depending on anilox line screen, as well as the smoothness of its surface, there are differences in the wear and life span of the doctor blade [8].

The role of the anilox roller is to precisely and consistently deliver the ink to the printing plate [9]. The engraving specification of the anilox roller is a key factor in achieving the highest possible opacity of the opaque white ink. The size and frequency of anilox cells control the volume of ink transferred to the printing plate [10]. The thickness of ink layer is directly connected to the speed of printing. At lower printing speeds, around 60 m/min, it is possible to achieve greater ink transfer and over 65% white ink opacity; however, the question of economic benefit can be posed. However, depending on the ink formulation, it may also mean that the increased printing speed will help "smooth" or "wet" the white ink.

The interactions between printing ink and printing substrate are influenced by properties of the printing ink, such as viscosity, surface tension, and wettability [11]. The surface free energy of the film is the basic parameter for good wettability of the substrate and its printing [12,13]. One of the key parameters that determine the print quality is the adhesion of the ink to the printing substrate [14]. The surface tension of the film significantly affects the white ink lay down and uniformity of the layer [15]. Treating the substrate prior to printing increases its surface tension. However, it has to be higher than the ink surface tension, so that the ink would adhere well to the substrate. Untreated films and foils have a low surface energy level and, as such, are not good for ink adhesion. Once the substrate has been treated, regardless of the type of treatment, any type of ink can adhere to it. This is especially important for printing with an opaque white color, which is usually printed first in sequence, although this is not always the rule. A print on which the ink does not adhere well to the substrate has a mottled appearance, that is, the ink film is not uniform [16]. Pinholes formation is mainly caused by bubbles in the white ink liquid when the roller supplement ink [17]. The abrasiveness of the substrate enables better ink adherence and improves ink wetting, and the printing is carried out without any issues [18].

The hardness of the polymer plate in combination with the selection of the mounting adhesive tape has a significant influence on the adherence of the ink and, thus, on the opacity of the opaque white ink [19]. This choice can affect the level of "mottle" effect and the appearance of pinholes. When printing full-tone (solid) colors, a less hard polymer plate is usually used in combination with a harder mounting tape. This enables a higher efficiency of ink transfer onto the substrate. The soft polymer plate tends to distribute the ink evenly over its surface, and the hard mounting tape firmly presses the polymer plate against the printing substrate, filling the voids on the surface and reducing the appearance of the "pinholing" effect. Furthermore, the smooth surface of the polymer cannot maintain a uniform ink layer thickness due to the surface energy of the plate. The abrasive surface of the polymer plate leads to an increase in white ink opacity; the abrasiveness can be achieved in several ways. The "old school" approach is based on solid colors rasterization using values in the 95–97% coverage range, and more recently, advanced screening technologies (Kodak HyperFlex, Esko Microcell, Mac Dermid LUX MicroCell). The mentioned technologies solve this by creating a texture on the surface of the plate that is extremely suitable for creating a uniform ink layer, and the texture itself is invisible on the print.

Furthermore, increasing the opacity of the opaque white ink can be achieved by using highly pigmented opaque inks [20]. Such inks are characterized by higher viscosity (particle size of the white titanium dioxide pigment is in a range of 200 to 400 nm), which affects the printing speed. However, with the development of new ink formulas, the viscosity of ink, including white ink, decreased. The new formulas are based on smaller pigment sizes and ink additives that improve printing properties.

Although a white underlayer is often required to achieve a high color saturation in attention-grabbing graphics, the cost of adding white ink can be significant, especially when the coverage of the graphic is large [21]. Therefore, the most important thing is to set the optimal acceptable opacity of the opaque white ink that can be easily managed while printing. Any increase in opacity usually requires an increase in the volume of the anilox roller, which ultimately results in increased ink consumption as well as an increase in total costs.

Generally, the 50% white ink opacity is acceptable in flexo-printing processes. Considering that the white ink makes for 40–50% of total ink cost, its optimization is very important, providing there is an acceptable level of printing quality. It is, therefore, important to analyze the extent to which the increase in the white ink opacity pays off and how it influences the improvement of the visual effect on a product. Savings in white ink expenses can significantly decrease the production costs. Furthermore, the question arises whether an ordinary observer could even visually detect the difference in color printed over white with higher ink opacity.

The primary objective of the research is the optimization of the production process, which includes the relationship between the white ink opacity and the print quality, and can significantly affect the consumption of white ink and, thereby, reduce total production costs. Optimization principles should aim to print the thinnest possible white ink layer that creates a sufficiently smooth surface for printing over it and, at the same time, ensures the acceptable opacity value. Therefore, it is necessary to determine by research how the thickness of the white ink layer affects the change in its opacity and the opacity of different color samples when printed on a transparent film. The analysis and interpretation of the results of the color samples opacity is based on their brightness, which complements the existing known theories and facts about the behavior of colors on the underlying white ink. The print quality of white ink is primarily based on the opacity value, but also on the uniformity of the ink laydown, which is determined by analyzing the pinholing effect that appears on the print. For the evaluation of the pinholing effect, new research methods were designed that provide a more complete insight into this issue.

## 2. Multi-Layer Packaging for Food Products

The research was carried out as part of designing a test print for multi-layer polymer flexible packaging for a food product Kulen, produced by Podravka, a company in Koprivnica, Croatia. Food product is packed in a thermoformed base on which the flexible packaging material, the subject of this research, is sealed.

The packaging material used for research in this paper consists of a multi-layered polymer material. Polymeric materials are commonly used for the production of food product packaging for several reasons: they are easy to manufacture and have excellent characteristics such as flexibility, strength, stiffness, and barrier properties, and they provide protection against unwanted microorganisms [22–24].

The composition of the packaging material from the research is the following: BOPET/PE/EVOH/PE (Figure 1). The upper external layer of the material that is printed on from the inside is a biaxially oriented polyester film polyethylene terephthalate (BOPET) that is 20 μm thick. The issues that occur in the process of printing polymer films are: poor wettability, lack of absorbency, low surface free energy, lack of dimensional stability during stretching, and temperature changes [25]. Due to the mentioned characteristics, most polymer films need to be pre-treated and prepared for the printing process. BOPET is used in various combinations with polymer or combined flexible packaging materials. In the printing process, it has high dimensional stability and resistance to solvents [26]. BOPET is characterized by good mechanical strength, resistance to tearing and stretching, high transparency, and gloss. It is, therefore, often chosen as a packaging material.

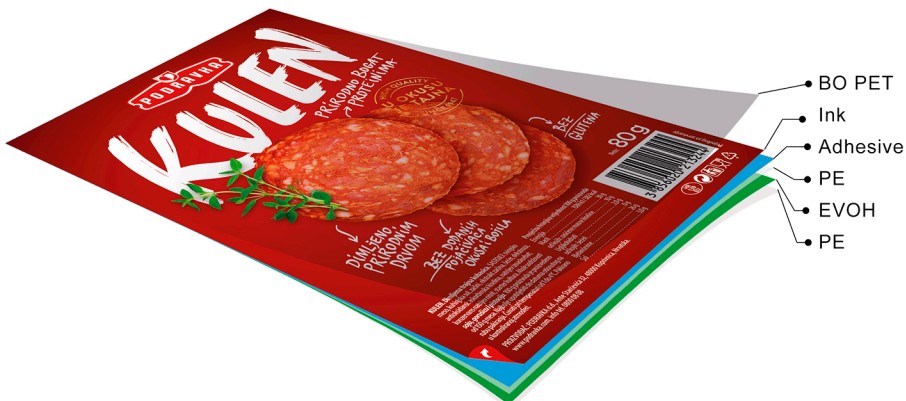

**Figure 1.** Layout of Kulen flexible packaging with an overview of individual material layers.

The following polymer layer in the mentioned packaging material is polyethylene (PE), i.e., PE/EVOH/PE, with a total thickness of 50 μm. It is one of the most important and present polymers in the production of food industry packaging. A layer of EVOH (ethylene vinyl alcohol) is placed in a sandwich between two layers of PE. It is a copolymer of ethylene and vinyl alcohol, and it serves as a barrier layer. Its role is primarily to protect the food product from the influence of oxygen and to retain the aroma and smell of the product. EVOH is moisture sensitive and has to be combined with materials that are hydrophobic, such as PE. In some applications of multi-layer flexible packaging, it can replace an aluminum layer. Integrating the EVOH layer into the packaging extends the shelf life of products sensitive to external influences. EVOH also has high mechanical strength, high resistance to oils and organic solvents, and high thermal stability [27].

## 3. Materials and Methods

This research was conducted as part of designing the test print for the mentioned flexible packaging. The research framework that describes the goal and process of the research is shown in Figure 2. By changing the anilox rollers, the volume of white ink increased in identical steps of 3 cm$^3$/m$^2$ throughout three stages of the printing experiment, while other parameters were constant, including the printing speed (200 m/min), the level of printing pressure (nip engagement: 75 μm for color inks and 150 μm for white ink), and the type of printing substrate. By increasing the volume of the anilox roller, the thickness of the ink layer on the print also changed, which further affected the characteristics of colors printed over white.

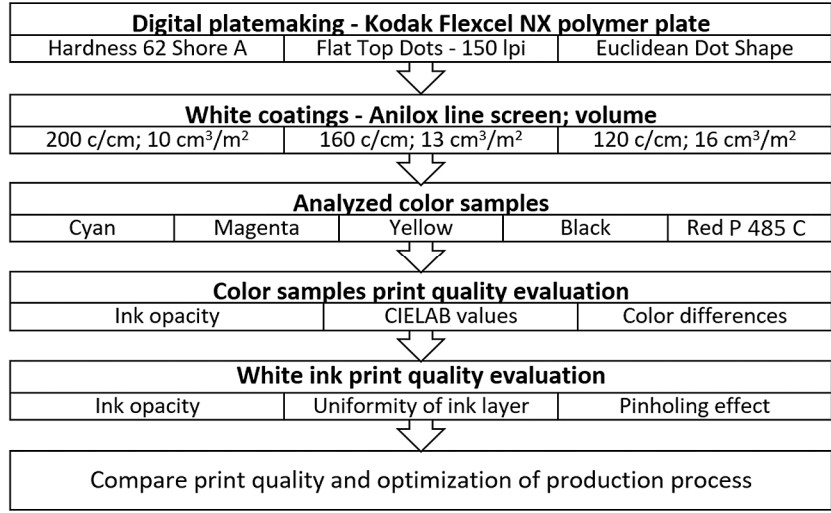

**Figure 2.** Research framework.

### 3.1. Digital Platemaking Process

The experimental part of this paper started with the design of a test form consisting of visual for flexible packaging of Podravka Kulen food products, and control wedges that will enable the evaluation of print quality, using acceptable and confirmed scientific methods and research techniques. The graphics preparation was carried out using professional PackZ software produced by Hybrid software. The design was derived from a total of seven separations: four process colors (CMYK), a spot red Pantone 485 C, an opaque white, and a partially applied matte varnish.

Cross-modulated (XM) screening technology, a combination of AM and FM technologies, was used in designing the printing plate. The flexographic photopolymer printing plate used for this research was Kodak Flexcel NX Digital Plate (hardness acc. to DIN 53505, Shore A is 62, plate thickness 1.14 mm). First, a thermal imaging film (thickness 0.165 mm) was exposed on the Flexcel NX Imager and was then laminated to the flexographic plate. After exposing the reverse and front side of plate, the film was removed. After main UV exposing, plates were processed with standard solvent process. A characteristic of Flexcel NX technology [28] for the production of photopolymer plates is the flattened top dot shape, which requires lighter "kiss" pressure in printing and, thus, enables a quality transfer of ink from the printing plate to the printing substrate [29].

### 3.2. Printing Process

UTECO ONYX 808 flexo central drum printing press with a web width of 1350 mm and a maximum repeat length in print [30] of 800 mm was used for test printing. Test samples were printed according to the "roll-to-roll" method at a printing speed of 200 m/min using solvent-based printing inks for flexible packaging. Reverse printing in a single pass was used, that is, a technique in which the ink is printed on the inside of the printing substrate. In this way, after the lamination process, the ink did not come into direct contact with the contents of the package, nor with the customers during the purchase, and it was not influenced by different weather conditions. In the second pass through printing machine to the outer surface of the printing substrate (so-called Surface printing), HUBER Gecko Frontal Uni matt varnish was printed. After the printing process, the two-component varnish needed seven days for polymerization and cross-linking under controlled conditions to be ready for lamination. In line with the halftone screen and defined minimum size of the halftone dot, the optimal anilox line screen for printing was determined. During the printing process, a substrate passed between plate cylinder and impression cylinder. The space between them must be optimal to give the proper printing pressure [31]. The substrate was polyester (polyethylene terephthalate-PET), chemically pretreated film on both sides during the production process, with surface energy of at least 42 dyn/cm. The printed film was laminated with PE/EVOH/PE film. Table 1 shows the detailed technical specifications of the prepress and printing process.

**Table 1.** Technical specifications of the prepress and printing process.

| Plate-Making Process Specification | |
| --- | --- |
| Ctp device | KODAK FLEXCEL NX Wide 5080 System |
| Resolution | 2400 dpi |
| Photopolymer type | Kodak Flexcel NX Digital Plate, 62 shore A |
| Plate thickness | 1.14 mm |
| Base layer thickness | 0.125 mm |
| Line ruling | 150 lpi (lines per inch) |
| Screening type | XM, Maxtone FX–Euclidean Dot Shape |

**Table 1.** *Cont.*

| Printing Specifications | |
| --- | --- |
| Flexo printing machine | Uteco ONYX 808 CI Flexo Printing Press |
| Printing speed | 200 m/min |
| Process/Spot flexo inks | HUBER Gecko Bond Top, Solvent based inks |
| White flexo ink | HUBER Gecko Xtreme White, Solvent based ink |
| Matt varnish | HUBER Gecko Frontal Uni |
| Printing width max. | 1200 mm |
| Plate mounting tape | 3M E315, 0.38 mm thickness |
| Print run | 75,000 labels |
| **Anilox Roller Specifications** | |
| Process color (CMYK) | 470 c/cm (1200 cpi); 3.7 $cm^3/m^2$ |
| Spot red P 485 C color | 160 c/cm (400 cpi); 13.0 $cm^3/m^2$ |
| Printing Viscosity–Process/Spot ink | 20–25 s DIN 4 |
| White ink–Stage I | 200 c/cm (500 cpi); 10.0 $cm^3/m^2$ |
| White ink–Stage II | 160 c/cm (400 cpi); 13.0 $cm^3/m^2$ |
| White ink–Stage III | 120 c/cm (300 cpi); 16.0 $cm^3/m^2$ |
| Printing Viscosity–White ink | 18–22 s DIN 4 |
| **Printing Substrate Specifications** | |
| Transparent film | BOPET Film Roll Polyester Film, Polyethylene Terephthalate; thickness: 20 μm |

*3.3. Evaluation*

The main goal of the research was to determine the print quality under the influence of different opaque white ink layer thickness. All measurements were performed on the printed PET film before the varnish printing and lamination stages. The measurement of opacity in the white ink layer (Figure 3) and color ink layer (Figure 4) on transparent (TRN) film was performed using a spherical spectrometer Xrite Ci64 (serial number 16,807), and was calculated based on CIE L*a*b* color space values. Technical specification of the spectrometer and measurement conditions: Measurement Geometrics d/8° (diffuse 8), Specular excluded, 4 mm measurement area/6.5 mm target window, Illuminant Type D50, and Standard Observer 2°. Diffuse 8 is a principle of geometry where the light source first hits the walls of a highly reflective coated sphere of the measurement device and this diffuse light illuminated the sample. The measurement was detected at 8° of the sample plane. According to ISO 6504-3, each measurement of samples requires three readings (over-black, over-white, and a measurement of the white backing) which results sample opacity percentage values. The opacity of the color sample is then measured as the difference of a measurement with black background and a measurement with white background taking into account the white backing [32]. The device reads the opacity values in the sample in the form of a percentage value between 0 (completely transparent) and 100 (completely opaque) [33], which follows human visual perception. One layer of uncoated smooth surface black paper board, with a grammage of 240 g/$m^2$ and thickness of 271 μm, was used as a black background during the measurement (L* = 24.9; a* = 1.5; b* = −0.8). Four-layer GMG Proof paper premium Semimatte, with a grammage of 250 g/$m^2$, was used as the white background (L* = 96.5; a* = −0.8; b* = −1.2).

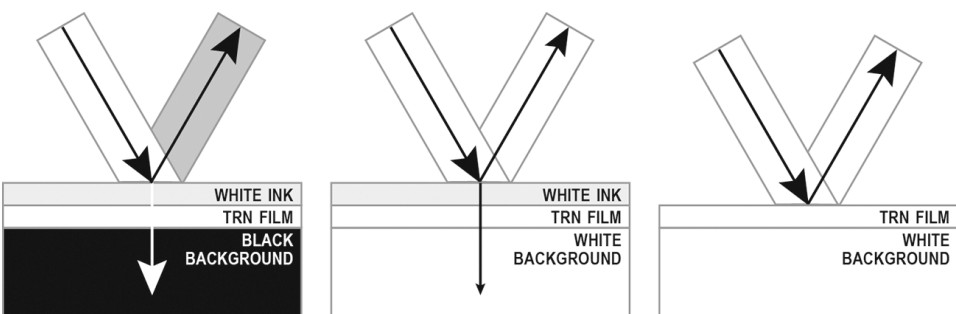

**Figure 3.** Measurement principle for white ink opacity.

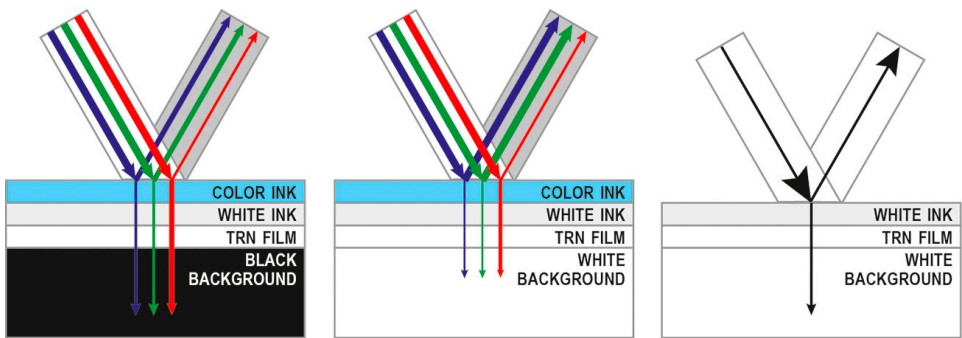

**Figure 4.** Measurement principle for color ink opacity.

Opacity (Equation (1)) is defined in terms of the contrast ratio, the ratio of the relative luminous reflectance of a coating placed over a black and a white substrate [34]:

$$O(\%) = \frac{R_b}{R_w} \times 100 \tag{1}$$

where:

$O$—Opacity;

$R_b$—Relative reflectance, black backing;

$R_w$—Relative reflectance, white backing.

CIELAB values of color samples based on Equation (2) for CIEDE2000 were used to calculate the color difference of the process colors and the spot red P 485 C color through the three stages of the experiment [35]:

$$\Delta E = \sqrt{\left(\frac{\Delta L'}{K_L S_L}\right)^2 + \left(\frac{\Delta C'}{K_C S_C}\right)^2 + \left(\frac{\Delta H'}{K_H S_H}\right)^2 + R_T \left(\frac{\Delta C'}{K_C S_C}\right)\left(\frac{\Delta H'}{K_H S_H}\right)} \tag{2}$$

where:

$K_L$, $K_C$, $K_H$—Correction factors related with observation environment;

$S_L$, $S_C$, $S_H$—Lightness, chroma, and hue weighting factors;

$R_T$—Rotation factor.

Table 2 shows subjective assessment metrics of color difference aligned with human perception and based on the CIEDE2000 equation. CIEDE2000 corresponds better with the way human observers perceive small color differences [36].

The set tolerance from Table 2 means the limit of the acceptable difference between the reference color and the sample, which will be applied in the presentation of the research results.

**Table 2.** Subjective assessment metric based on CIEDE2000 Color difference [35].

| k | $\Delta E_{min}(k)$ | $\Delta E_{max}(k)$ | Perception of Color Differences |
|---|---|---|---|
| 1 | 0.0 | 0.5 | Hardly |
| 2 | 0.5 | 1.5 | Slight |
| 3 | 1.5 | 3.0 | Noticeable |
| 4 | 3.0 | 6.0 | Appreciable |
| 5 | 6.0 | 12.0 | Much |
| 6 | 12.0 | 24.0 | Very much |
| 7 | 24.0 | $\infty$ | Strongly |

In order to evaluate the quality of the underlying white ink on a transparent film, it is not enough to only determine its opacity. For years, the opacity was used as the printing standard of white ink. However, the ink layer uniformity and the appearance of pinholing on the print is also reflected in the quality of the white ink. By increasing the volume of the anilox roller, more ink can be transferred to the printing substrate, thereby creating a thicker layer and increasing the opacity of the white ink. This is still not enough for a high-quality white ink layer, which impacts the high-quality flexographic print visual experience [37]. Therefore, to determine the quality of the white ink laid down on the printing substrate, in addition to measuring the opacity, the uniformity of the ink layer on the print should be analyzed. This includes the evaluation of surface smoothness and pinholing effect.

In order to analyze the uniformity of the white ink layer on the print, it is necessary to capture the selected areas on the prints in all three stages of the experiment, considering the variation in the anilox roller volume. For this purpose, the Digital Microscope Dino-Lite AM4000 was used. It had a resolution of 1.3 megapixels and a built-in LED light that enabled a better view of the object that was being captured. Analysis samples were captured with a magnification of 200× and a resolution of 1280 × 1024 pixels. ImageJ 1.47 software was used for processing and analysis of microscopic images, and various image analysis techniques were used for evaluation. The measurement results must be presented in real values (mm, μm, ...). Therefore, when measuring, it is necessary to set a corresponding size scale based on known values at identical magnification. The scale was set to 512:1 for image analysis, i.e., 512 pixels amount to 1 mm.

## 4. Results and Discussion

### 4.1. Analysis of White Ink and Color Ink Opacity

The results of measuring white ink opacity and selected color samples' opacity are shown in Table 3, and were obtained as the average value of five measurements. The printing experiment took place in three stages that had constant printing conditions, thus enabling the investigation of a variable parameter, which was the volume of the anilox rollers. The first stage used an anilox roller with a volume of 10 cm$^3$/m$^2$, and it was increased by 3 cm$^3$/m$^2$ through the remaining two stages of the experiment.

**Table 3.** Ink opacity throughout three stages of the experiment with different anilox volume.

| | Opacity (%) | | |
|---|---|---|---|
| Color Samples | Anilox Roller Volume (cm$^3$/m$^2$) | | |
| | 10 | 13 | 16 |
| C | 51.53 | 53.46 | 57.8 |
| M | 48.52 | 50.35 | 54.68 |
| Y | 44.45 | 45.4 | 49.99 |
| K | 60.23 | 61.1 | 67.35 |
| R | 50.55 | 52.4 | 56.77 |
| W | 42.08 | 44.33 | 49.47 |

The increase in the white ink opacity (Table 3) was directly related to the anilox roller volume for white color, while the increase in the opacity of the color samples related to the white ink opacity.

The diagram from Figure 5 primarily shows the change in the white ink opacity (W) due to the increase in the ink layer, which resulted from the increase in the anilox roller volume. Anilox roller volume of process colors (CMYK) and spot red P 485 C color (marked as R in tables and figures) was constant throughout all three printing stages, which means that their relative opacity also did not change. However, as the opacity of the opaque white ink increased, the absolute opacity of the process colors and the spot red P 485 C color increased. Generally speaking, yellow as the lightest color had the lowest opacity, and black as the darkest color had the highest opacity. This effect can be attributed to the physical fact that yellow has the lowest blocking power of all inks. As a result, yellow is more dependent on a white undercoat to block background colors (in this case, a black backing) than other colors.

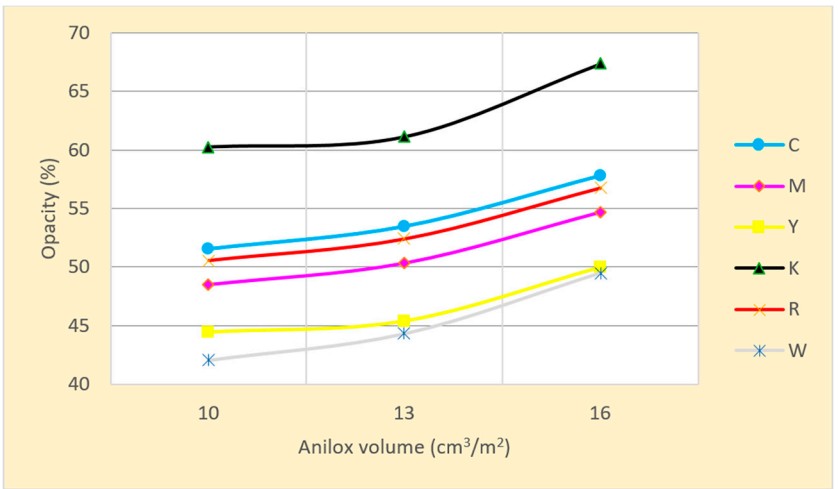

**Figure 5.** Change in the opacity of the white ink and colored samples throughout the three stages.

A linear increase in the volume of the anilox roller by 3 cm$^3$/m$^2$ (2 BCM) during the experiment did not result in a linear increase in the white ink opacity. The curve was slightly exponential. Increasing the volume from 10 cm$^3$/m$^2$ (O = 42.08%) to 13 cm$^3$/m$^2$ resulted in an increase by 2% (O = 44.33%) in opacity, while a further increase to 16 cm$^3$/m$^2$ caused an increase in opacity by 5% (O = 49.47%). The curves of the tested color samples show a similar trend in increasing opacity. However, this increase was less exponential compared to white ink.

Figure 6 shows the absolute increase in opacity based on the linear increase in the volume of the anilox roller in the last two stages of the experiment in relation to the reference values from the previous stages of the experiment.

The diagram in Figure 6 shows that the total white ink opacity in the third stage of the experiment significantly increased the opacity of the colored samples (around 4.5%) compared to the second stage of the experiment (about 1–2%). This proved that for every linear ink film increase, the opacity increased exponentially (the increase in the third stage of the experiment was more than twice as big, as in the second stage of the experiment). According to the authors Valdec et al., a further increase in ink layers increased the opacity value, but only up to a certain point, after which it began to decrease [38]. The results can be linked to lightness, i.e., the amount of light in individual color samples (Figure 7). The increase in white ink layer in the second stage of the experiment had a significantly smaller effect on light (yellow, L = 89.8%) and dark (black, L = 36.3%) colors compared to the remaining three color samples of medium lightness (L = 47–60%). By further increasing the ink layers in the third stage of the experiment, the opacity values were more uniform with a positive deviation of black.

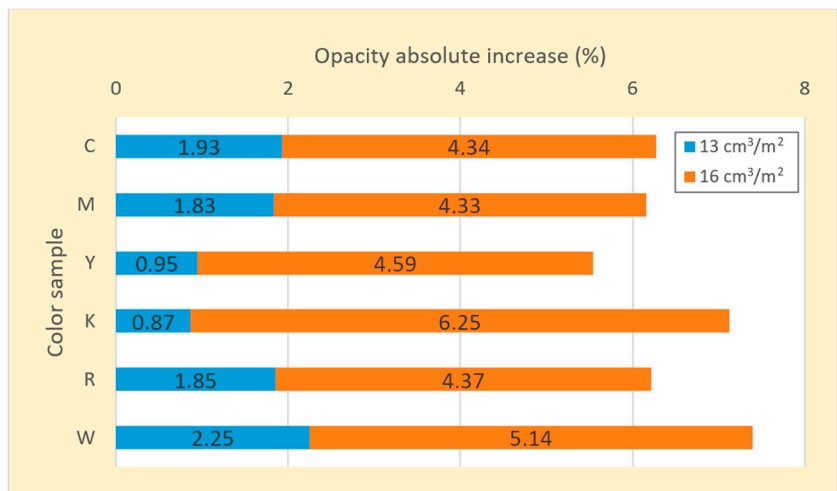

**Figure 6.** Absolute increase in the opacity of the white ink and colored samples.

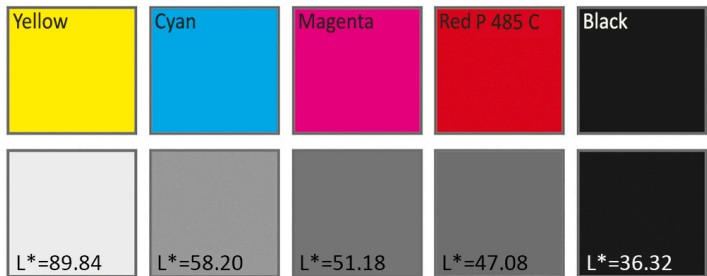

**Figure 7.** Color samples and their achromatic versions, arranged according to the amount of light in the color.

However, the absolute increased opacity values did not give us a real insight into the change in opacity, with respect to the values from the previous stage of the experiment. Therefore, it was necessary to determine the percentage deviations for each individual color sample, which showed the relative change in relation to the previously observed experiment stage (Figure 8). The relative deviation in opacity showed a very similar behavior of cyan, magenta, and red P 485 C samples due to the increase in white ink film throughout the last two stages of the experiment.

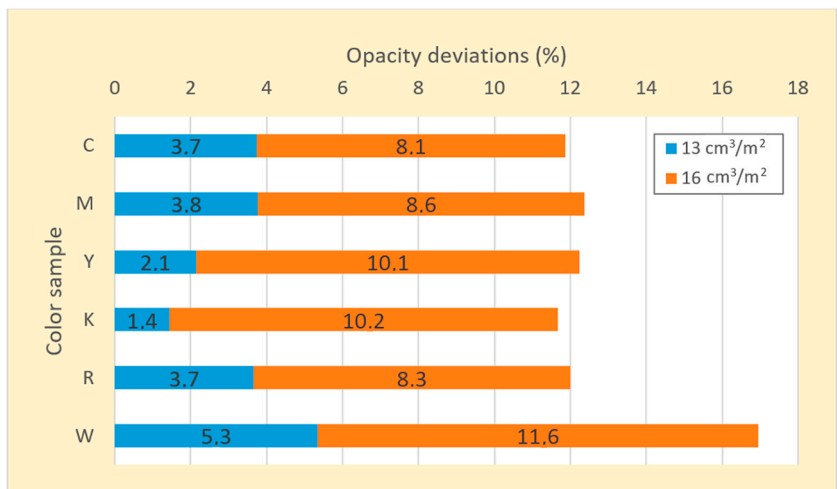

**Figure 8.** Relative opacity deviation for white ink and color samples at two last stages.

The deviation amounted to 4% after the first stage of the experiment. After the second stage, it was twice as big and amounted to 8%. The lightest and darkest samples, yellow and black, had the smallest relative deviation after the second stage (about 2%), and the largest relative deviation after the third stage (about 10%). The graph also shows an identical overall relative increase in opacity for all samples, which amounted to about 12%.

### 4.2. Analysis of Colorimetric Values in Color Samples

The influence of anilox roller volume on the colorimetric values of the process colors and the spot red P 485 C color is shown using the color differences. The color difference was determined based on the CIELAB value of color samples measured on the white background, and calculated using Equation (2) for CIEDE2000 ($K_L = K_C = K_H = 1$). The values of the DE2000 color differences and difference in lightness ($\Delta L$) for the last two stages of the experiment, compared to the initial state from the first stage of the experiment for all tested samples, can be found in Table 4.

**Table 4.** Color differences $\Delta E2000$ and differences in lightness $\Delta L$ for all tested color samples.

| Color Samples | Anilox Roller Volume (cm$^3$/m$^2$) | | | |
|---|---|---|---|---|
| | Stage II: 13 | | Stage III: 16 | |
| | $\Delta L$ | $\Delta E2000$ | $\Delta L$ | $\Delta E2000$ |
| C | 0.36 | 0.489 | 2.35 | 2.448 |
| M | 0.18 | 0.838 | 1.92 | 2.043 |
| Y | 0.51 | 0.712 | 2.82 | 2.694 |
| K | −0.73 | 0.931 | −2.23 | 2.306 |
| R | 0.36 | 0.841 | 1.76 | 2.246 |

Generally speaking, the values of color differences in the second stage of the experiment compared to the initial state were in the range of $\Delta E2000 = 0.5$–$1.0$. According to the classification of color differences (Table 2), this means that despite the higher white ink film thickness on printouts, the observer could hardly or very slightly visually distinguish the same color samples [39]. The values of the color differences in the third stage of the experiment were significantly higher ($\Delta E2000 = 2.0$–$2.7$), which was consistent with the increase in color samples ink opacity. According to the classification mentioned, the color differences were noticeable to an ordinary observer. This means that a significant increase ink layer (6 cm$^3$/m$^2$/4 BCM) was necessary for an ordinary observer to notice the final color difference. Yellow and cyan showed more sensitivity to an increase in anilox volume for white ink than other chromatic colors. This effect can be attributed to their greater brightness than magenta and the red spot color. By increasing the white ink layer, all tested color samples generally became lighter ($\Delta L > 0$), except for black, which became darker ($\Delta L < 0$). This can be connected with the greater transparency of chromatic inks compared to black ink, which still had higher opaque with equal ink film thickness.

Figure 9 graphically shows CIEDE2000 color differences for all color samples in the two last stages of the experiment compared to the initial state where an anilox roller volume of 10 cm$^3$/m$^2$ was applied.

According to the results of the research, a significantly thicker film of white ink is needed, which would result in a significant increase in opacity in order to visually notice the color difference on printouts. However, the question of the efficiency and economy of applying thicker white ink film arises. A better insight into the cost-effectiveness of increasing the ink film thickness can be shown through the total costs of white ink, especially for larger print runs.

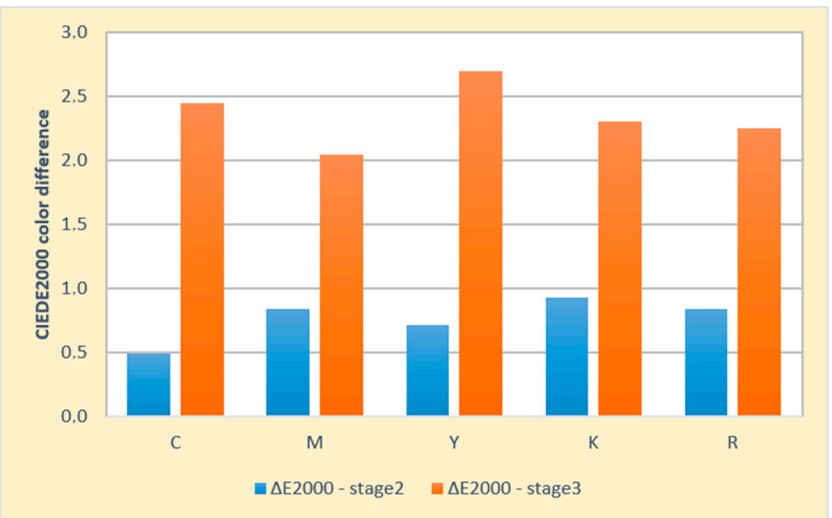

**Figure 9.** Color difference at two last stages of the experiment compared to the initial state.

### 4.3. Print Quality Analysis of White Ink

The print quality of white ink throughout the three stages of the experiment, apart from measuring opacity (Section 4.1), was evaluated based on the uniformity of ink layer on the printouts and the appearance of pinholes [40]. Microscopic images captured on printouts were analyzed through ImageJ software (tool: Interactive 3D Surface Plot) that transformed density into proportional height to see just how thick the ink film was [41]. During the transformation process, the "Invert" setting was selected, so that the lightest elements, which represent the thickest ink film, would be displayed as relatively high.

The projection of ink density in the three stages of the experiment clearly showed greater uniformity of the surface as the thickness of the ink film increased, which is visible in Figure 10 starting from left to right. Also, the three samples differed significantly regarding the number and size of pinholes on the print. The far right sample had a significantly smoother surface structure and no visible pinholes, as in the previous two samples. Various factors can cause the appearance of pinholes, and one of the most important ones is the ink film thickness.

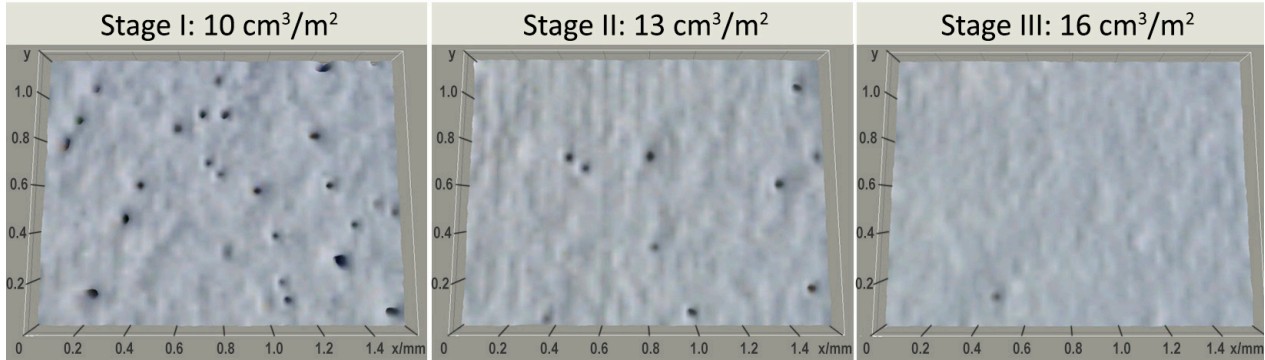

**Figure 10.** Topographic view of white ink film thickness at different printing stages.

Using ImageJ software (tool: Analyze Particles), the area and the number of pinholes that appeared on the solid color of white were measured. When evaluating the area and the number of pinholes, the color threshold method of image processing was used. Thresholding is a method where the image converts from color or grayscale mode into a binary image mode. The red mask that was thus created in the image was used to measure the area of the pinholes. The following values of color threshold in HSB color space were used: hue 0–255; saturation 80–255; brightness 0–140.

Measurement results of area and the number of pinholes at three stages of printing are shown in Table 5. The results were grouped according to pinhole sizes, and the holes with a surface area of less than 100 $\mu m^2$ were not taken into account, since, due to their extremely small size, they do not significantly affect the print quality. In order to compare, that would be the area of one laser dot (10 $\mu m$ dot size at resolution of 2540 dpi) and it is also extremely difficult to reproduce it on the printing plate.

**Table 5.** Number of pinholes at three stages of the experiment.

| | Number of Pinholes | | |
| --- | --- | --- | --- |
| Pinhole Area ($\mu m^2$) | Anilox Roller Volume ($cm^3/m^2$) | | |
| | 10 | 13 | 16 |
| 100–199 | 34 | 9 | 1 |
| 200–299 | 18 | 1 | 0 |
| 300–399 | 11 | 1 | 0 |
| 400–499 | 3 | 0 | 0 |
| 500–999 | 6 | 1 | 0 |
| 1000 and more | 7 | 0 | 0 |

The pinholes usually do not have a regular shape, but for the sake of better understanding the size of these pinholes and in order to compare them, further analysis used more appropriate values. Pinholes over 400 $\mu m^2$ (20 $\mu m$ dot size), and especially those over 900 (30 $\mu m$ dot size), had a significant impact on print quality. Based on the results, the print obtained with an anilox roller volume of 13 $cm^3/m^2$ met the criteria of absence of large pinholes on the impression. However, this does not mean that this choice of anilox roller is acceptable. The opacity of the white ink should also be taken into account for the final selection.

## 5. Conclusions

When printing transparent substrates, a white ink underlayer is required to reproduce saturated colors. The white ink has a pronounced effect on color perception that is printed over it. The change in opacity of color samples is related to their brightness. According to the conducted research, the increase in white ink opacity from 42% to 44% in the second stage of the experiment was very slightly visually noticeable in color samples on printouts ($\Delta E2000 = 0.5-1.0$). This means that 1.3 times the amount of white ink was needed without any improvement in the final results. However, a visually noticeable change in color samples ($\Delta E2000 = 2.0-2.7$) occurred due to the increase in white ink opacity from 42% to 49% in the third stage of the experiment (1.6 times larger amount of white ink was needed). This can significantly increase the production costs of the final product. On average, an increase in opacity of 1% requires an increase in anilox roller volume of 1.2 $cm^3/m^2$.

Medium lightness colors (cyan, magenta, and red P 485 C) more strongly reacted to changes in white ink opacity compared to light (yellow) and dark (black) colors. It can be concluded that light colors require a higher opacity of the white ink layer in order to comply with the defined standards and look the most faithfully possible on the print. Therefore, it is not possible to speak generally about the acceptable value of white ink opacity in print, since it significantly depends on the color scheme that was applied in the product design.

Also, the distribution of the white ink on the printing substrate significantly impacted the quality of the print. A thicker white ink layer had a more uniform distribution of the ink, which also meant a smoother surface and a less pronounced pinholing effect. Therefore, such a surface reflected more light and looked lighter. Also, colors printed over white had significantly better characteristics. All color samples were lighter compared to the referent state ($\Delta L = 1.7-2.8$); only the black color was darker ($\Delta L = -2.2$). Furthermore, by increasing the volume of the anilox roller for white ink, the lightness increased more in the lightest color samples. For example, there was a greater increase in yellow ($\Delta E = 2.7$)

compared to magenta ($\Delta E = 2.0$). However, a thicker white ink layer can create additional issues in printing, as well as issues with the smell of packaging material.

The white ink can have extremely high opacity, but if the ink layer is not uniform, and has a significant number of pinholes on the print, this can contribute to creating a bad visual effect. Therefore, in order to achieve high print quality, the pinholing effect must be taken into account, in addition to high white ink opacity. The evaluation of qualitative reproduction parameters gave important indicators that can significantly improve the production process and result in increased quality.

**Author Contributions:** Project administration, supervision and conceptualization, R.T. and D.V. (Dean Valdec); investigation and methodology, D.V. (Dean Valdec) and M.T.; resources, R.T.; visualization and validation, D.V. (Damir Vusić); writing—review and editing, R.T., D.V. (Dean Valdec), M.T. and D.V. (Damir Vusić). All authors have read and agreed to the published version of the manuscript.

**Funding:** This research received no external funding.

**Institutional Review Board Statement:** Not applicable.

**Informed Consent Statement:** Not applicable.

**Data Availability Statement:** Measurement data are available per request by corresponding author email.

**Conflicts of Interest:** The authors declare no conflict of interest.

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
