# Peer review of "The Impact of Underlying Opaque White Coating Parameters on Flexographic Print Quality"

_applsci, doi:10.3390/app13158575_

Round 1

Reviewer 1 Report

Comments to the Author
The authors of the manuscript applsci-2423095-peer-review-v1, investigated the impact of anilox roller volume of the opacity of opaque white, as well as the opacity and CIELAB values of colors which are printed over it.  Authors used anilox roller with variable volumes, while other experimental conditions were kept constant. They used microscopic with ImageJ software to analysis the printed images. The topic is interested in researchers working in basic print research as well as the print technology. The manuscript layout is clear, informative, the experimental data are explained, discussed, and well presented. Thus, I consider the article borderline acceptable from this angle. Thus, I recommend the manuscript for publication in the Journal in its current form.

Author Response

Dear reviewer,
We appreciate the time and effort you devoted as a reviewer to our manuscript.
Thanks for the positive comments.
New version of the article has also been uploaded.
Best regards
Authors

Reviewer 2 Report

Opaque white ink plays a critical role in printing on transparent substrates, providing coverage for packaging content while allowing the printing of other colors as specifications. This study investigated the influence of anilox roller volume on the opacity of opaque white ink and its impact on the opacity and CIELAB values of colors printed over it. The research was conducted in three stages, progressively increasing the volume of the anilox roller in linear increments. The findings reveal an exponential relationship between the volume of the anilox roller and both white ink opacity and the opacity of color samples. It also presented the adverse impact of the pinhole due to insufficient volume. I would like to recommend accepting it for publication after further improvement.

1.      There are several former works especially tacked the white ink in flexography: (a) Study on Quality Evaluation and Optimization Scheme of White Ink in Flexography, Innovative Technologies for Printing and Packaging, 2022; (b) Research on the Covering Property of White Ink Applied on Transfer Paperboard, Innovative Technologies for Printing and Packaging, 2016; (c) Comparison of Colorimetric Values of Prints Made with Cyan Ink on Different Polymer Materials, Procedia Engineering, 2014.

2.      The introduction should clearly articulate the motivation behind the study.

3.      It would be better to add a more quantitative summary of the results and discussion in the introduction to clarify the novelty. Also, the addition of one or two sentences indicating potential uses of the results and contributions of this study will be helpful for readers.

4.      In the introduction, the author states that viscosity is a key factor in printing performance. However, there is no mention of the viscosities of the white and color inks used in the study. Including this information would help readers understand the specific parameters involved in the experiments.

5.      It is better to show the optical figures of visual appearance in the opacity of the white ink and colored samples throughout the three stages.

6.      Please add the scale bar in Figure 10.

7.    The format of the references needs to be revised.

The quality of the English language in the provided text is generally good. The sentences are well-structured and effectively convey the ideas. However, the use of tense is inconsistent and should be revised. 

Author Response

Dear reviewer,
We appreciate the time and effort you devoted as a reviewer to our manuscript, and the insightful comments to improve the article.
We have managed to incorporate changes that reflect most of your suggestions.
More detailed actions that have been taken are described in the attached document, and a new version of the article has also been uploaded.
Best regards
Authors

Reviewer 3 Report

The review report on the manuscript titled "The Impact of Opaque White Coating Parameters on Flexographic Print Quality" is very interesting, especially in the field of graphic technology where such topics are always welcome since there are not many of them.

Therefore, I recommend acceptation of this MS after the authors consider the following.

1.       The authors should highlight the novelty of the paper.

2.       If possible, I suggest reducing the number of self-quotations and instead, where possible, including some new ones.

3.       Why is DeltaC CMC used for calculation instead of CIE Delta E 2000, despite the fact that you emphasized in lines 203-204 that CIE Delta E 2000 better corresponds to the way human observers perceive small differences?

If you chose to use CIE Delta E 2000, why is ΔE written in Table 4? It should be consistent across all tables.

4.    In Figure 7. L need stars L*

I recommend having the paper reviewed and polished by a professional English language editor to ensure its fluency and grammatical accuracy.

Author Response

(The authors gave the same response as above.)

Reviewer 4 Report

The article investigates the impact of anilox roller volume on the opacity of opaque white ink used for printing on transparent substrates. The study demonstrates that increasing the anilox roller volume leads to exponential changes in both white ink opacity and the opacity of colours printed over it.

The article is well-designed, as is the research methodology. However, there are some shortcomings to this study that need to be addressed or explained. 

Firstly, I would suggest adding the word *underlying* to the Title of the paper such as: The Impact of Underlying Opaque White Coating Parameters on Flexographic Print Quality

There are some formal mistakes: 

·         line 102: The composition of the packaging material from the research is the following: BO PET / PEevoh / PE (Figure 1). Some slash (/) marks are missing, and EVOH is written in lowercase letters. I believe it should be corrected to:  BO/PET/PE/EVOH/PE.

·         line 161: When describing the surface energy of polymer films, it is correct to use the unit "dyn/cm" (dynes per centimetre) instead of dynes (please correct 42 dynes to 42 dyn/cm).

Lines 113-116: Can you explain why EVOH film is sandwiched between two PE layers? Does it improve the relatively poor barrier properties of PE? Is this material a functional barrier against potential migrants from solvent-based printing inks?

At the beginning of the Materials and Methods section, you provide a description of the experiment primarily using a schematic representation of the research process steps. Such representation may be clear to someone in the field of graphic technology, but for researchers unfamiliar with printing terminology and methods, it might be helpful to provide a brief additional explanation of the process, particularly regarding the evaluation of printed samples.

Lines 148-157: At what point the underlying white opaque ink is printed? Is everything printed (white coating + CMYK and Pantone Red) in a single pass? Furthermore, what is the composition of matte varnish – is it also solvent based? How does this affect the possible invisible set-off migration (for FCM)?

Lines 174-190 – can you give more explanation why you need to consider the white backing when measuring ink opacity. And why this value isn’t included in Equation 1 - where reflectance measured over black backing is divided with reflectance measured over white backing?  

Lines 195 – 206: The DeltaECMC algorithm was used to calculate the colour difference between the measured CIELAB values of the printed colour samples. However, for subjective assessment metrics (Table 2), you referenced the CIE deltaE2000 algorithm instead. Since the equations for DeltaECMC and CIE deltaE2000 are not equal and were developed by different organizations, why didn't you use the CIE deltaE2000 equation in the first place for the evaluation of your results? It is known that the very first version of DeltaE, DeltaE 1976, was the Euclidean distance of colours in Lab. However, as it turned out that Lab was not as uniform in perception as once thought, the algorithm was revised in 1984 (CMC), 1994, and finally in 2000, resulting in the most accurate and most complicated Lab-based DeltaE algorithm to date which corresponds better with the way how human observers perceive small colour differences.

I assume that the criterion stated in Table 2 (which seems to resemble the deltaE 1976 (deltaEab) algorithm rather than the DeltaE2000 equation) may not differ significantly in the interpretation of the deltaECMC algorithm. However, it would still be necessary to provide accurate criteria, if available. Furthermore, it would be helpful to provide a clearer definition of an acceptable result for the printing and packaging industry in the Table 2.
Please check following literature references for the interpretation of deltaE2000 results: Yang, Y.; Ming, J.; Yu, N. Color Image Quality Assessment Based on CIEDE2000. Adv. Multimed. 2012, 2012, 273723 https://www.hindawi.com/journals/am/2012/273723/  and Kumar, M. Interdisciplinarnost Barve 2.Del.; Jeler, S., Kumar, M., Eds.; Društvo koloristov Slovenije: Maribor, Slovenia, 2003; pp. 89–100.

Lines 238-239 Can you please emphasize in the Results section what is the known acceptable opacity of the white ink underlayer for the packaging industry when printing on flexible materials?

In the study, you have demonstrated that a higher application of white ink results in a more uniform layer without pinholes. However, what is the economic viability of using a larger ink volume in practical printing? Additionally, you mention issues regarding the odour of the print and the impact of packaging on the sensory properties of packaged food. Are there any inks available that do not pose such issues (e.g., low migration inks)?

There are about 18% self-citations in the paper (7 out of 39 cited references), while the usual standard allows up to 10% self-citations. Authors should consider removing unnecessary self-citations wherever possible.

Author Response

(The authors gave the same response as above.)

Author Response

(The authors gave the same response as above.)

Round 2

Reviewer 4 Report

I would like to thank the authors for their detailed explanations regarding the part of the manuscript that was not clearly written and for considering most of my comments. With these improvements, the manuscript can go into the publication process as far as I am concerned.

Author Response

Thank you very much for the positive review, but also for your effort to adequately improve the article.

Thank you very much for your cooperation

Authors